# Probing defect dynamics in monolayer MoS$_2$ via noise nanospectroscopy

Seung Hyun Song[1,2], Min-Kyu Joo[1,2], Michael Neumann[1,2], Hyun Kim[1,3] & Young Hee Lee [1,3]

Monolayer molybdenum disulfide (MoS$_2$) has received intense interest as a strong candidate for next-generation electronics. However, the observed electrical properties of monolayer MoS$_2$ exhibit several anomalies: samples universally exhibit unexpectedly low mobilities, n-type characteristics, and large contact resistances regardless of contact metal work function. These anomalies have been attributed to the presence of defects, but the mechanism behind this link has been elusive. Here we report the ionization dynamics of sulfur monovacancy defects in monolayer MoS$_2$ probed via noise nanospectroscopy, realized by combining noise–current analysis with atomic force microscopy. Due to the nanoscale dimension of the in situ channel defined by the tip size, we probe a few monovacancy defects at a time. Monovacancy defects exhibit switching between three distinct ionization configurations, corresponding to charge states 0, −1, and −2. The most probable charge configurations are 0 and −1, providing a plausible mechanism to explain the observed anomalies of MoS$_2$ monolayers.

[1] Center for Integrated Nanostructure Physics, Institute for Basic Science (IBS), Suwon 16419, Republic of Korea. [2] Sungkyunkwan University (SKKU), Suwon 16419, Republic of Korea. [3] Department of Energy Science, Sungkyunkwan University (SKKU), Suwon 16419, Republic of Korea. Seung Hyun Song, Min-Kyu Joo and Michael Neumann contributed equally to this work. Correspondence and requests for materials should be addressed to Y.H.L. (email: leeyoung@skku.edu)

The structural defects that preexist in two-dimensional transition metal dichalcogenide materials, or are introduced during sample processing, strongly influence the physical properties. In the case of $MoS_2$, several anomalous observations have been attributed to the presence of defects: electrical transport measurements on thin sheets almost universally reveal n-type characteristics[1–5]; contact resistances are large, varying only weakly even for large differences in contact metal work function[1,6–8]; and even the highest measured mobilities of monolayer $MoS_2$ are unexpectedly low, $<150$ cm$^2$ V$^{-1}$ s$^{-1}$ (refs. [3,5]). Furthermore, higher defect densities in $MoS_2$ monolayers are correlated with a degradation of sample mobility[4].

Previous work studying the microscopic structure and energy states of defects in $MoS_2$ has identified the sulfur monovacancy ($V_S$) as the most abundant defect type[4,9–11] with reported densities ranging between $4 \times 10^{10}$ and $5 \times 10^{13}$ cm$^{-2}$ (refs. [4,6,9,12–14]). This insight comes from theoretical calculations, as well as from high-resolution transmission electron microscopy (TEM) and scanning tunneling microcopy (STM). In contrast, much less is known about the dynamics of defects in $MoS_2$. Low-frequency (LF) noise measurements, traditionally used to study the defect dynamics of silicon-based field effect transistors (FETs)[15–17], have identified scattering by charged impurities as the dominant noise source in monolayer $MoS_2$ FETs[18]. However, the FET sample geometry imposes a lateral channel dimension of several µm; a channel area of $1 \times 1$ µm$^2$ contains $\sim 10^4$ defects, and thus data acquired in this geometry may not represent the properties of individual defects. Moreover, defects associated with the insulating oxide substrate and electrode contacts represent additional noise sources that further complicate the analysis.

In this study, we probe the defect dynamics of monolayer $MoS_2$ by combining LF noise measurements with conductive atomic force microscopy (C-AFM), thereby extending LF noise spectroscopy into the nanoscale regime[19].

## Results

**Probing defect charge states by noise spectroscopy.** Our experimental C-AFM setup (Fig. 1) probes the electrical properties of $MoS_2$ monolayers grown by chemical vapor deposition (CVD) and deposited on a gold substrate. The metallic AFM tip and the substrate function as the source and drain electrodes, respectively, and the small area in which the AFM tip contacts the $MoS_2$ surface represents the device channel. Thus, the channel dimension $A$ is shrunk to $A \ll \pi r^2$, with tip radius $r < 25$ nm, corresponding to 1–8 individual monovacancies for the defect density $4 \times 10^{11}$ cm$^{-2}$ previously determined for our $MoS_2$

material[13]. Since only a few defects contribute to the acquired signal, spectra collected in this fashion more closely represent individual defect characteristics. Furthermore, the simplified geometry of a metal/semiconductor/metal tunnel junction eliminates the complications associated with the FET geometry.

For defects that have multiple discrete charge states, LF noise arises from switching processes occurring between these states. We utilize LF noise measurements as a direct probe of the defect dynamics of the sulfur monovacancy $V_S$. In the neutral state $V_S(0)$, the vacancy (Fig. 2a) is occupied by two electrons, corresponding to the valence of the missing sulfur atom, S$^{2-}$. Theoretical work has predicted the presence of sulfur monovacancies in $MoS_2$ to result in the appearance of new states ($a_1$, $e$) within the bandgap, as illustrated in Fig. 2b (refs. [9,10,20]). By trapping electrons in the $e$-state, the neutral defect can ionize into different charge states. Theoretical studies[10,11,21] have predicted that depending on the position of the Fermi energy $E_F$ within the energy bandgap of $MoS_2$, the neutral state $V_S(0)$, the charged state $V_S(-1)$, and possibly $V_S(-2)$ (refs. [20,21]) are most stable, while other charge states are unstable due to strong electrostatic repulsion. The relative stability of $V_S$ charge states is shown in the energy diagram in Fig. 2c (adapted from refs. [10,11]).

We summarize the key results of our study in Fig. 2d. In noise spectra from tunneling current measurements performed on our $MoS_2$ samples, we observe two distinct switching processes that occur between the three vacancy defect states $V_S(0)$, $V_S(-1)$, and $V_S(-2)$. The switching process $V_S(0) \leftrightarrow V_S(-1)$ is far more likely than the process $V_S(-1) \leftrightarrow V_S(-2)$. The stable $V_S(0)$ and $V_S(-1)$ states are separated by a relatively high kinetic barrier; in contrast, the kinetic barrier for the metastable configuration $V_S(-2)$ is much lower. (Theoretical work[10,11,20,21] is divided on the question whether $V_S(-2)$ can be the ground state for any value of $E_F$ within the bandgap; refs. [10,11] predict that it cannot. In that scenario, however, the configuration $V_S(-2)$ can be transiently occupied nonetheless, as observed in our work.)

**Noise spectral analysis.** We now proceed to describing the results of our measurements on monolayer $MoS_2$/Au samples in detail. Figure 3a, c show representative C-AFM ac-current traces, acquired at room temperature, which are dominated by switching between discrete levels (extremes of switching indicated as red lines), reminiscent of the random telegraph signal characteristic for a noise source with discrete states. The corresponding noise power spectra $S_I(f)$ (Fig. 3b, d) are described well as sums of two Lorentzian noise functions (Eq. (1)) with characteristic frequencies $f_\alpha$ and $f_\beta$ (green and cyan lines, respectively), demonstrating that two separate trapping/recombination processes

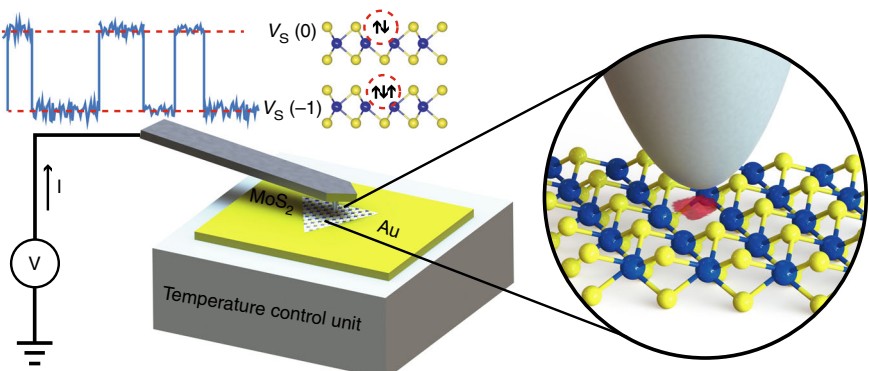

**Fig. 1** Schematic illustration of LF noise measurement system. $MoS_2$ flakes transferred to a conducting gold-coated substrate are probed by conductive AFM. The current flowing through the sample, $I = I_{bias} + I_{ac}$, is composed of the bias current $I_{bias}$ that is constant due to the 100 MΩ series resistor, and the noise signal $I_{ac}$

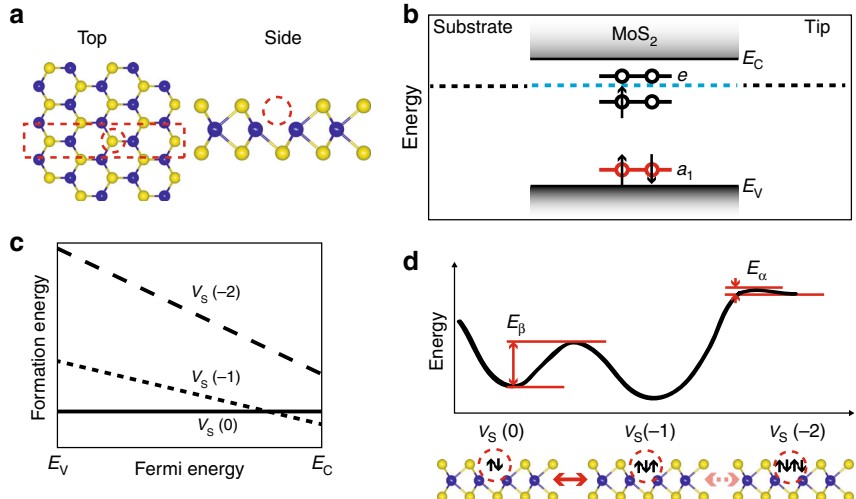

**Fig. 2** Structure and energy diagram of MoS$_2$ monolayer sample. **a** Illustration of MoS$_2$ lattice structure, with a sulfur monovacancy; top view, side view. Theoretical work has shown the spatial extent of the vacancy's electronic orbitals to be ~2 unit cells[10,32]. **b** Schematic bandstructure diagram of the sample. The presence of a vacancy defect $V_S$ leads to the formation of mid-gap states ($a_1$, $e$) (refs. [9,10,20]). Here the configuration $V_S(-1)$ is shown. The Fermi level $E_F$ lies within the MoS$_2$ energy bandgap; since in our sample geometry, MoS$_2$ is in contact with a gold substrate, $E_F$ is fixed by the work function of gold, at a position close to the conduction band minimum $E_C$. **c** Schematic stability diagram of the charge states in monolayer MoS$_2$, as a function of $E_F$ (adapted from refs. [10,11]). **d** Bottom: electron configurations of $V_S$, in the charge states 0, −1, and −2. Top: schematic energy diagram of these $V_S$ charge states constructed from our experimental observations. Electron capture/detrapping events result in switching between discrete states, $V_S(0) \leftrightarrow V_S(-1)$ and $V_S(-1) \leftrightarrow V_S(-2)$. The energy barriers $E_\beta$ and $E_\alpha$ determine the degree of meta-stability of $V_S(0)$ and $V_S(-2)$, respectively. $E_V$, valence band maximum

associated with well-defined energies occur in our sample. In contrast, our measured noise spectra exclude the possibility of a dominant flicker noise component[15,22], which would result in a contribution proportional to $1/f$ (black line). (See Supplementary Note 3 for further discussion of spectral shapes.)

We obtain the numerical values of the relevant physical parameters via least-squares fits to the sum of two Lorentzian functions

$$S_I(f) = \frac{A_\alpha(I_{\text{bias}})I_{\text{bias}}^2}{1 + \left(\frac{f}{f_\alpha}\right)^2} + \frac{A_\beta(I_{\text{bias}})I_{\text{bias}}^2}{1 + \left(\frac{f}{f_\beta}\right)^2} + C, \quad (1)$$

where $I_{\text{bias}}$ denotes the preset dc-bias current amplitude, $A_\alpha$ and $A_\beta$ are the amplitudes of the Lorentzian functions of widths at $f_\alpha$ and $f_\beta$, and $C$ represents a flat background describing the measured system noise spectrum (magenta line in Fig. 3b, d; for a discussion of the noise floor and related issues, see Supplementary Methods and Supplementary Fig. 3). The characteristic LF noise frequencies are $f_\alpha \simeq 4$ Hz and $f_\beta \simeq 1$ kHz; these two frequencies reflect the slow switching processes (Supplementary Fig. 2) and fast switching processes (Fig. 3a, c) observed in the current traces. The two frequencies $f_\alpha$ and $f_\beta$ do not vary appreciably as a function of bias current (Supplementary Fig. 4), justifying the use of the Shockley–Read–Hall model in the following. In contrast, the noise power amplitudes $A_\alpha$, $A_\beta$ vary dramatically as a function of bias current (see discussion in Supplementary Notes 1 and 2).

The numerical values of $f_\alpha$ and $f_\beta$ will depend on the electron capture cross-sections, and on the height of kinetic barriers between these discrete states. In order to disentangle the separate roles of these physical quantities, we perform C-AFM measurements as a function of temperature, over the range 150–300 K. The temperature-dependent charging/discharging time constant $\tau(T)$ of defect states, in the low injection limit, is modeled by the Shockley–Read–Hall equation,

$$\tau(T) = \frac{1}{2\pi f(T)} = \frac{1}{\sigma^{(0)} \times \exp\left(\frac{-E}{k_B T}\right) \times N_T \times \nu}, \quad (2)$$

with the high-temperature limit capture cross-section $\sigma^{(0)}$, the kinetic barrier $E$ between discrete states, the trap state density per unit volume $N_T$, the electron velocity $\nu$, and absolute temperature $T$. Figure 3e, f present the temperature dependence of the characteristic frequencies $f_\alpha$ and $f_\beta$, in the coordinates $\log(f)$ vs $1000/T$ that reflect the form of Eq. (2). Qualitatively, $f_\alpha \simeq 4$ Hz exhibits no observable change as a function of temperature. In contrast, the evolution of $f_\beta$ indicates an exponential decrease upon cooling; such an exponential temperature dependence has been previously reported in tunneling junctions where the electron velocity is independent of temperature[23,24].

Using Eq. (2), we quantitatively estimate the defect states' activation energies and capture cross-sections from the slopes and extrapolated $f$-axis intercepts in Fig. 3e, f. The activation energies corresponding to the characteristic frequencies $f_\alpha$ and $f_\beta$ are $E_\alpha < 15$ meV and $E_\beta = 23$ meV (error bound 10–43 meV), respectively. The ratio of scattering cross-sections calculated from $f$-axis intercepts is $\sigma_\beta^{(0)}/\sigma_\alpha^{(0)} \simeq 830$. With the previously determined areal density of $V_S$ defects in our sample, $(N_T)^{2/3} = (4 \pm 1) \times 10^{11}$ cm$^{-2}$ (ref. [13]), and a tunneling electron velocity known to be $\nu = 10^7 - 10^8$ cm s$^{-1}$ (ref. [16]), we estimate the absolute scattering cross-section values to be of the order $\sigma_\alpha^{(0)} \approx 10^{-24} - 10^{-23}$ cm$^2$ and $\sigma_\beta^{(0)} \approx 10^{-21} - 10^{-20}$ cm$^2$. The observation that $\sigma_\alpha^{(0)} \ll \sigma_\beta^{(0)}$, by three orders of magnitude, suggests that the two capturing processes involve dissimilar charge states; electron capture by neutral defects is far more efficient than by negatively charged sites. Moreover, the magnitudes of $\sigma_\alpha^{(0)}$ and $\sigma_\beta^{(0)}$ fall within the typical ranges of electron capture cross-sections of negatively charged and neutral sites, respectively[16].

The spatial uniformity of our measured noise spectra, each representing an average over the probing area defined by the tip size, implies that at any position at least a single vacancy resides

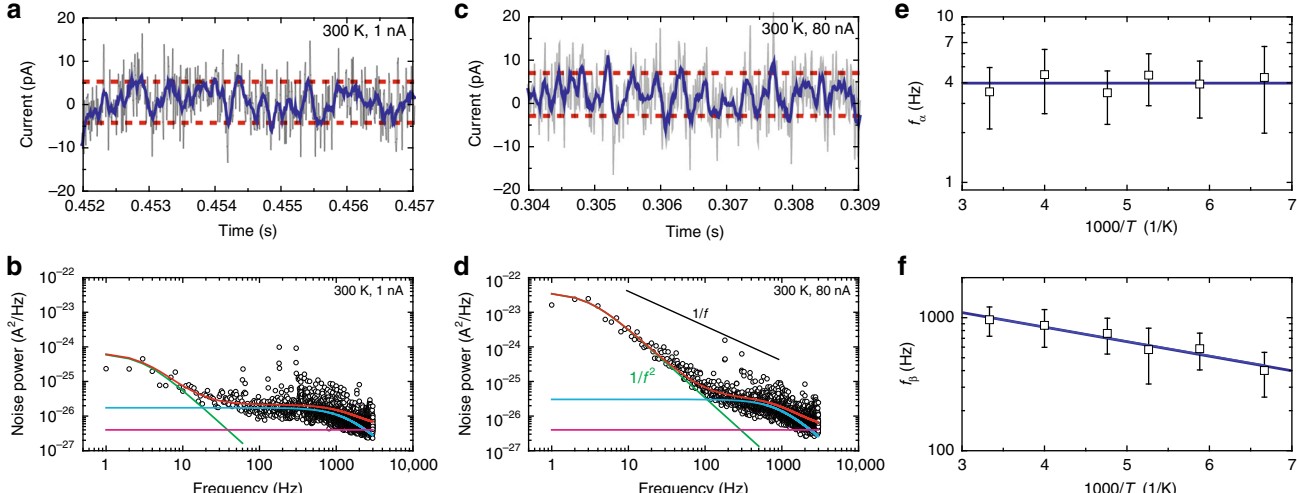

**Fig. 3** LF noise–current data and associated spectra. **a** Noise signal current trace ($I_{ac} = I - I_{bias}$, for $I_{bias} = 1$ nA) acquired at room temperature (gray line); low-pass filtered current trace (blue line). The current trace exhibits switching between multiple discrete levels involving a small number of vacancy defects (red lines highlight the extremes of switching). As low-pass filter, we use a Savitzky-Golay filter of polynomial order 3. **b** Noise power spectrum $S_I(f)$ corresponding to the data set in **a**. **c, d** Corresponding data acquired on the same sample at $I_{bias} = 80$ nA. Measured spectra are well described by Eq. (1), comprising two Lorentzian functions of widths $f_\alpha = 4$ Hz (green line) and $f_\beta = 1$ kHz (cyan line). In contrast, spectra are inconsistent with $1/f$ behavior expected for flicker noise (black line). Measurement sensitivity limit is set by instrument noise (violet line). **e, f** Temperature dependence of characteristic frequencies, (**e**) $f_\alpha$ and (**f**) $f_\beta$, in coordinates log($f$) vs. $1000/T$. Via Eq. (2), the fitted slopes (blue lines) allow us to evaluate the kinetic barriers associated with switching processes. We observe no temperature dependence of $f_\alpha$, establishing an upper limit of $E_\alpha < 15$ meV for the process $V_S(-1) \leftrightarrow V_S(-2)$. For $f_\beta$, we obtain the kinetic barrier $E_\beta = 23$ meV for the process $V_S(0) \leftrightarrow V_S(-1)$. Error bars, standard deviation

in the AFM tip/MoS$_2$ contact area. For a probing area $A \ll \pi r^2$, with tip radius $r < 25$ nm, we can place a lower bound $\rho \gg 5 \times 10^{10}$ cm$^{-2}$ on the density of defects, consistent with earlier characterization of our MoS$_2$ material[13].

## Discussion

The schematic energy diagram presented in Fig. 2d summarizes the inferred charge state dynamics: our observation of two well-separated Lorentzian peaks in measured power spectra reveals that two distinct switching processes are present, between three discrete charge states of $V_S$. Specifically, the high value of cross-section $\sigma_\beta^{(0)}$ indicates the involvement of a neutral electron trap, $V_S(0)$; in contrast, the low value of $\sigma_\alpha^{(0)}$ is consistent with a transition between two negatively charged states, $V_S(-1)$ and $V_S(-2)$. In addition to the kinetic barriers quantified above, the switching time constants determined from current traces allow us to estimate the relative energy levels of the charge states. The fast switching process occurring at rate $f_\beta$ (Fig. 3a, c) reveals approximately equal durations spent in its high-current state $V_S(0)$ and its low-current state $V_S(-1)$, indicating that these two charge states occupy levels of similar energy. In contrast, slow switching at rate $f_\alpha$ (Supplementary Fig. 2) exhibits asymmetric time constants, staying mostly in its high-current state $V_S(-1)$, indicating that its low-current state $V_S(-2)$ is a higher energy state. This is in agreement with the theoretical prediction that $V_S(-2)$ is not the ground state[10,11]. (See Supplementary Note 4 for a more detailed discussion of the energy diagram shown in Fig. 2d.)

We note that the preceding analysis does not consider the interaction between sulfur monovacancies. This is justified by the large average distance between $V_S$ sites; the defect density $(4 \pm 1) \times 10^{11}$ cm$^{-2}$ corresponds to an average distance 16–21 nm, i.e., 51–65 unit cells.

The energy diagram shown in Fig. 2d is consistent with previous theoretical work. According to theoretical calculations, the two charge states $V_S(0)$ and $V_S(-1)$ are expected to be nearly

equi-energetic if the Fermi level is close to the conduction band minimum (Fig. 2c), corresponding to $E_F \approx -4.8$ eV, with respect to the vacuum level[11]. This condition is indeed fulfilled: in our sample, monolayer MoS$_2$ is in intimate contact with a metallic gold substrate. Due to the much higher density of states of the metal, the Fermi level of the MoS$_2$ monolayer will follow the work function of thermally evaporated gold, which is known to be 4.7–4.9 eV (refs. [25,26]) (see Supplementary Note 7 for discussion).

We now discuss the implications of our findings for the case of monolayer MoS$_2$ FET devices. In that situation, the coexistence of neutral and negative charge states in monolayer MoS$_2$ would result in a locally inhomogeneous surface potential, leading to charge scattering and a degradation of carrier mobility, consistent with the relatively low mobilities experimentally found in monolayer MoS$_2$ (refs. [3,5]). Separately, the presence of sulfur vacancies has been linked to the experimental observations that contact resistances of MoS$_2$ devices are high due to Fermi level pinning[1,6–8,21], and that thin MoS$_2$ samples almost universally exhibit n-type carrier transport. This attribution to sulfur vacancies should be regarded as tentative in the case of exfoliated natural MoS$_2$ samples: natural MoS$_2$ can exhibit variable stoichiometry even within samples, including regions of sulfur excess[6,27], as well as a variety of impurity atoms that could lead to doping[27] (see Supplementary Notes 5 and 6 for an extended discussion). For CVD-grown MoS$_2$ monolayers, however, these complications are absent, and the realization of the $V_S(-1)$ state in a significant fraction of the abundant defects would account for all of the anomalies observed in MoS$_2$ FET devices.

The above reasoning implies that sulfur monovacancies have a tendency to ionize into their $V_S(-1)$ state, and in a more speculative extrapolation of our work, we suggest that existing theoretical work supports such a scenario: if a conduction band electron is trapped at the $e$-state of a monovacancy, forming the configuration $V_S(-1)$, the local chemical potential is raised to the energy of the $e$-state, located close to the conduction band minimum $E_C$ (Fig. 2b). In turn, theoretical work[10,11] has shown that the closer the chemical potential is located to $E_C$, the more

the $V_S(-1)$ state is favored, as shown in the stability diagram in Fig. 2c; this would result in a local self-stabilization of the $V_S(-1)$ configuration. Accordingly, the presence of a sufficiently high density of sulfur monovacancies exhibiting a tendency toward $V_S(-1)$ formation and a locally raised chemical potential would stabilize the overall Fermi level $E_F$ at a position closer to $E_C$, which would explain the almost universally observed n-type behavior of $MoS_2$, as well as the Fermi level pinning that underlies the high contact resistance of $MoS_2$. (On the possible objection to this argument that $V_S$ is an acceptor and thus cannot lead to n-type behavior[11], we remark that this objection assumes that the number of electrons in the sample is conserved; this however is not the case in FET devices, nor in our sample.)

In summary, combining the high spatial resolution of C-AFM with LF noise spectroscopy provides a powerful means to quantify defects and explore their dynamics. The high sensitivity of our technique to the presence of sulfur vacancies—detecting defect levels on the order $10^{11}$ cm$^{-2}$, corresponding to 0.002% of sulfur atoms, or a local stoichiometry of $MoS_{1.99996}$—suggests that it is a tool complementary to other characterization techniques such as TEM and STM that are commonly utilized to quantify $MoS_2$ defect densities.

Most significantly, our results represent the first experimental observation of charge state switching in sulfur monovacancies in $MoS_2$, and our work adds significant detail to the understanding of the dynamic behavior of these defects. We find that three monovacancy charge states are accessed, $V_S(0)$, $V_S(-1)$, and $V_S(-2)$; furthermore, in good agreement with theoretical predictions, we observe that a high percentage of $V_S$ defects are in negative charge states, and combined with previous theoretical work, our observations suggest a plausible explanation for the unexpectedly low mobility, high contact resistance, and peculiar n-type behavior of monolayer $MoS_2$.

Looking forward, the experimental approach explored in our work presents a powerful means to attain greater understanding of the defect dynamics in a wide range of two-dimensional materials, assisting in endeavors to push the limits imposed by defects on device performance, or to exploit the properties of defects in novel device concepts.

## Methods

**Sample preparation.** Monolayer $MoS_2$ samples are grown on $SiO_2$ substrates by CVD, using ammonium heptamolybdate as precursor[28]. The gold substrate is prepared by thermal deposition of 5 nm Cr and 50 nm Au layers. Prior to transferring the $MoS_2$ monolayer sheet onto this substrate, it is submersed in trichloroethylene (TCE) for 30 min to fully remove organic surface contaminants. For the transfer of the $MoS_2$ monolayer sheet, we use the following wet transfer procedure: the CVD-grown $MoS_2$ flakes on $SiO_2$ are coated with PMMA (A$_4$, 2000 r.p.m., 60 s) and baked at 160 °C for 1 min. The sample is then floated in 1 M KOH solution kept at 80 °C, in order to delaminate the PMMA/$MoS_2$ film from $SiO_2$. Subsequently, PMMA/$MoS_2$ are transferred to a deionized water bath to rinse off residual potassium ions, and the film is transferred onto the gold-coated substrate. Following this transfer, the sample is submerged in TCE for 30 min to remove PMMA.

**Sample characterization.** We use Raman spectroscopy and ac-mode AFM to verify that $MoS_2$ sheet samples prepared in this manner consist of monolayers, and to confirm sample quality and uniformity. Supplementary Figure 1b presents Raman spectra collected at three different positions on the sample discussed in this article. The Stokes shifts of the primary Raman-active phonon modes $E_{2g}$ (386 cm$^{-1}$) and $A_{1g}$ (404.5 cm$^{-1}$) are consistent with those reported for monolayer $MoS_2$ in the literature[29]; Raman spectra are identical across the flake. Raman spectra are collected on a WITec alpha-300 microscope, using a $100\times$ objective and a Nd:YAG-laser of wavelength 532 nm, at a power of 0.5 mW; ac-mode AFM data are acquired on a Park Systems XE-7 AFM, using silicon cantilevers of spring constant 40 N/m and tip radius $r < 10$ nm.

**Conductive atomic force microscopy setup.** Figure 1 depicts the LF noise measurement setup. Samples are mounted inside the vacuum chamber of a Hitachi

E-Sweep AFM, where they are clamped onto a temperature control unit consisting of a Joule heater and a cold finger connected to a liquid nitrogen bath. Samples are annealed at 80 °C in vacuum for 2 h prior to measurements; all measurements are performed in high vacuum ($\sim 5 \times 10^{-6}$ torr). We perform C-AFM measurements using an ElectriMulti75 AFM cantilever with a Pt-coated tip of radius $r < 25$ nm, using a contact force $\sim 20$ nN. We use the internal voltage source of Hitachi E-Sweep to supply a dc-bias voltage between the AFM tip and the gold substrate. The resulting current is sensed through a noise measurement system consisting of a series resistor (100 MΩ), current preamplifier (SR570, Stanford Research Systems), and a data acquisition system (National Instruments DAQ-4431) that records both the dc current as well as its ac fluctuations, at an acquisition rate 100 kHz (ref. [30]). The series resistor prevents the AFM tip from overheating by keeping the current below 100 nA.

**Crystal structure visualization.** We use the software VESTA to draw crystal structure schematics[31].

**Data availability**. The data that support the findings of this study are available from the corresponding author upon request.

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

## Acknowledgements

This work was supported by IBS-R011-D1.

## Author contributions

S.H.S. and M.-K.J. conceived the study. H.K. and S.H.S. fabricated the samples. S.H.S, M.-K.J., and M.N. prepared the experimental apparatus. S.H.S. and M.-K.J. performed the experiments. S.H.S., M.-K.J., and M.N. evaluated and interpreted data, and wrote the manuscript under the guidance of Y.H.L. as primary investigator. All authors discussed the results and commented on the manuscript.

## Additional information

**Competing interests:** The authors declare no competing financial interests.

