## [Peer Review File · Nature Communications]

Reviewers' comments:

Reviewer #1 (Remarks to the Author):

S. H. Song and his co-workers reported the investigation of sulfur monovacancy defects in single-layer MoS₂ by using conductive atomic force microscopy (C-AFM). Since sulfur vacancies are the most common point defects in MoS₂, the detailed examination of their electronic properties have great significance for potential applications. The applied experimental method in the paper (combining LF noise measurements with conductive C-AFM) is quite unique and the main results about the distinct charge states appear to be solid. However, the discussion of the results is not very convincing from theoretical point of view, and I think the general conclusions are not supported sufficiently by the experiments. In my viewpoint, the present form of the manuscript is not suitable for publication in Nature Communications. Some detailed comments and questions are listed as follows:

1, The authors argued that in their sample the $V_s(0)$ and $V_s(-1)$ defects have similar energies due to the Fermi level shift close to the conduction band minimum (CBM). The origin of this Fermi level shift is the presence of the gold substrate which sounds reasonable. However, the general physical mechanism presented in the discussion part is not clear. The argument about the Fermi level shift caused by electron trapping and the "self-stabilization" of the $V_s(-1)$ is doubtful. Note that the Fermi level determines the $V_s(0)$ and $V_s(-1)$ concentrations not vice versa. Overall, I think the present experimental results explain the low mobility and n-type behavior of the MoS₂ samples where the Fermi level is close to the CBM (like in the case of gold substrate), but still not give explanation of the universal n-type behavior measured on different substrates.

2, In the measurements great spatial uniformity was observed of the spectra across the MoS₂ flakes. However, previous STM measurements (ACS Nano 8, 2880–2888 (2014), see ref 6) highlight Fermi level shift by 1 eV over tens of nanometers in spatial resolution. This doping is surface and defect-chemistry-related which certainly affects the charged state of the sulfur vacancy. For instance, in a recent STM measurement (Sci. Rep. 6, 29726 (2016)) both neutral ("a1" state) and charged ("e" state) sulfur vacancies were measured on gold substrate in a more direct method showing also spatial inhomogeneity. What could be the reason of the measured spatial uniformity in the present work?

3, Related to the Fermi level position; in Fig 2b the electron occupancy is confusing. The figure shows the neutral $V_s(0)$ situation, while the Fermi level is higher than the "e" defect state. It is worth noting that in the case of the charged state $V_s(-1)$ the defect levels are also modified (see ref 13).

4, Usually the C-AFM measurements are sensitive to the AFM tip. How many AFM tip (and sample) was used during the measurements?

Reviewer #2 (Remarks to the Author):

The authors present here a work on probing defect dynamics in monolayer MoS₂ via noise nanospectroscopy. Using a conductive atomic force microscope (C-AFM), they have studied the sulfur vacancies in a MoS₂ monolayer deposited on gold. In particular, they have performed noise measurements to probe the different charge states of the sulfur vacancy and characterize its dynamics, in particular the stability of the first charge state. By doing so, they provide a potential explanation for the reduced carrier mobility as well as the n-doping in MoS₂.

The major claims of the paper are the influence of the sulfur vacancies (which seem to be dominant in MoS₂ but without excluding other possibilities) on the charge mobility reduction in

MoS₂ and overall, the stability of the first charged state (one trapped electron), and its role in the n-doping of MoS₂. This latter constitutes a real novelty in the field and is probably of high interest. This study could however be extended to other type of defects in MoS₂ in order to control or modulate the electronic behavior.

The experiments and their corresponding interpretations presented here are rather convincing, in particular with respect to the noise measurements. However, despite a brief explanation in the supplementary material, I think that an STM image of the defect considered here, i.e. the sulfur vacancy, compared to previous experimental determination and/or theoretical modelling would be to my opinion more convincing for the reader. Indeed, if the case of oxygen can be a priori discarded, many other possibilities exist in terms of defects in a MoS₂ monolayer. For example, Mo substitution of Sulfur can occur, which is more complicated to detect and would create very reactive dangling bonds able to trap electrons. This latter aspect among other defects should also be considered for the global explanation of n-doping in MoS₂ which I think the authors have reduced too quickly to the single case of sulfur vacancy. Therefore, the discussion should be opened by the authors for more general possibilities beyond the present specific case of sulfur vacancy.

As a conclusion, I think the findings presented here are very interesting, and give a promising interpretation of MoS₂ n-doping, but a more accurate characterization of the defect is required and a discussion on the general possibilities of n-doping with several defects beyond the specific system considered here should be open before going further in the acceptance process for publication in Nature Communication.

Reviewer #3 (Remarks to the Author):

The manuscript presents a study of the ionization dynamics of sulphur vacancies in monolayer MoS₂ probed via noise nanospectroscopy and realized by using noise-current analysis in the conductive AFM mode. The manuscript concludes on existence of 3 distinct ionization configurations corresponding to different charge states. The results are interpreted for explanation of previously observed electrical deficiency of MoS₂ based devices. The noise experiment is thoroughly conducted and the manuscript is well written. Although the work is interesting, it is rather incomplete. Somewhat global conclusions are derived from a minimal amount of experimental results. Whereas all the drawn conclusions seem to be relatively plausible, the present work relies on a single experiment (Fig 3) with a very few parameters to change (i.e. 2 bias currents and a narrow temperature range). Moreover, the manuscript does not present a clear direct experiment, unambiguously linking the measured noise spectrum with the proposed electrical state of the vacancies. Thus, the statement 'noise measurements as a direct probe of the defect dynamics of the sulfur monovacancy' remains unproven and far-fetched.

More specifically:

- It would be interesting to investigate a different defective state of the sample, for example after inducing additional defects, e.g. by ion irradiation.
- Fig 1S (b, c) is largely inconclusive. It is impossible even to derive the flake thickness out of it. Part (c) doesn't bring any new information. It would be essential to perform conductive AFM mapping on both large and small scales and at least try to derive different electrical states of defects from there. Additionally, it should be compared with the surface potential distribution over the same area, ideally using a calibrated setup, which is relatively easy for the gold substrate.
- Fig 2: How original is the Figure? It seems that many parts are borrowed from Refs. 9, 12, 13.
- Fig. 2d: How were the energy profiles assigned to certain defect states? Is it purely schematics? Additionally, it says: 'The energy diagram shown in Fig. 2d is consistent with previous theoretical work.' My understanding that it is built up based on previous works and fits well to the conclusions of the present paper. The authors should be more specific on how Fig 2d was derived.
- How does the presence of the gold substrate affect the dynamics of the defects (i.e. by modifying E_f in respect to the defect level)?
- It would be useful if the authors could elaborate more on relative independence of both

frequencies (f_a and f_b) on current.

- I should be clarified how assignment of certain frequencies/capture cross-sections to specific charge defects was performed.
- 'two well-separated Lorentzian peaks in measured power spectra': I suppose the authors mean Fig 3 b,d. It is not obvious that the peaks can be accurately fitted by Lorentzians.
- 'Specifically, the high value of cross section (0) indicates the involvement of a neutral electron trap, $VS(0)$; in contrast, the low value of (0) is consistent with a transition between two negatively charged states, $VS(-1)$ and $VS(-2)$.' : this statement is rather hand waving. It is surely a possible explanation, but not confirmed by a direct experiment.
- Also, considerations on work function are again based on possible scenarios and data taken from literature rather than own measurements. There is no direct prove.

We thank the referees for their constructive and helpful comments. In response to these comments, we have revised several sections of our manuscript in order to clarify the points flagged by the referees. We also made significant additions to the Supplementary Information, where we discuss a number of more complex issues in detail. In the marked version of the manuscript and SI, we have highlighted these amended or added passages in blue.

Reviewer #1

General comment: "Since sulfur vacancies are the most common point defects in MoS₂, the detailed examination of their electronic properties have great significance for potential applications. The applied experimental method in the paper (combining LF noise measurements with conductive C-AFM) is quite unique and the main results about the distinct charge states appear to be solid."

We thank the referee for this positive evaluation.

Comment #1: "However, the general physical mechanism presented in the discussion part is not clear. The argument about the Fermi level shift caused by electron trapping and the "self-stabilization" of the V_S(-1) is doubtful. Note that the Fermi level determines the V_S(0) and V_S(-1) concentrations not vice versa. Overall, I think the present experimental results explain the low mobility and n-type behavior of the MoS₂ samples where the Fermi level is close to the CBM (like in the case of gold substrate), but still not give explanation of the universal n-type behavior measured on different substrates."

We thank the referee for pointing out that the description of our proposed mechanism of V_S(-1) stabilization in the case of MoS₂ FET devices was not sufficiently clear. We have now elaborated that the proposed V_S(-1) self-stabilization is a *local* effect, i.e., if capture of an electron by V_S(0) leads to formation of V_S(-1), the raising of the local chemical potential to the *e*-level locally stabilizes the V_S(-1) state, based on the theoretical work of Noh *et al.* (2014) and Komsa *et al.* (2015). If a sufficiently high density of V_S are present, the locally raised chemical potentials will also lead to a raise in the overall Fermi level towards the CBM. We quote the amended passage (page 10 of the manuscript):

"The above reasoning implies that sulfur monovacancies have a tendency to ionize into their V_S(-1) state, and in a more speculative extrapolation of our work we suggest that existing theoretical work supports such a scenario: if a conduction band electron is trapped at the *e*-state of a monovacancy, forming the configuration V_S(-1), the *local* chemical potential is raised to the energy of the *e*-state, located close to the conduction band minimum E_C (see Fig. 2b). In turn, theoretical work^{10,11} has shown that the closer the chemical potential is located to E_C , the more the V_S(-1) state is favoured, as shown in the stability diagram in Fig. 2c; this would result in a local self-stabilization of the V_S(-1) configuration. Accordingly, the presence of a sufficiently high density of sulfur monovacancies exhibiting a tendency towards V_S(-1) formation and a locally raised chemical potential would stabilize the overall Fermi level E_F at a position closer to E_C , which would explain the almost universally observed n-type behaviour of MoS₂, as well as the Fermi level pinning that underlies the high contact resistance of MoS₂."

If, by this mechanism, a significant fraction of monovacancies in a MoS₂ sample is in the V_S(-1) configuration, n-type behaviour and strong Fermi level pinning would follow as a consequence,

consistent with the widely reported experimental correlation between these phenomena and the presence of sulfur vacancies.

In our revised manuscript, we have now explicitly restricted this discussion to CVD-MoS₂, since in exfoliated natural MoS₂ a large variety of other defects complicates the picture. The relevant passage (page 10 second paragraph) is quoted below.

“Separately, the presence of sulfur vacancies has been linked to the experimental observations that contact resistances of MoS₂ devices are high due to Fermi level pinning^{1,6-8,22}, and that thin MoS₂ samples almost universally exhibits n-type carrier transport. This attribution to sulfur vacancies should be regarded as tentative in the case of exfoliated natural MoS₂ samples: natural MoS₂ can exhibit variable stoichiometry even within samples, including regions of sulfur excess^{6,28}, as well as a variety of impurity atoms that could lead to doping²⁸ (see Supplementary Notes 11 and 14 for an extended discussion). For CVD-grown MoS₂ monolayers, however, these complications are absent, and the realization of the V_S(-1) state in a significant fraction of the abundant V_S defects would account for all of the anomalies observed in MoS₂ FET devices.”

Comment #2: "In the measurements great spatial uniformity was observed of the spectra across the MoS₂ flakes. However, previous STM measurements (ACS Nano 8, 2880–2888 (2014), see ref 6) highlight Fermi level shift by 1 eV over tens of nanometers in spatial resolution. This doping is surface and defect-chemistry-related which certainly affects the charged state of the sulfur vacancy. For instance, in a recent STM measurement (Sci. Rep. 6, 29726 (2016)) both neutral (“a1” state) and charged (“e” state) sulfur vacancies were measured on gold substrate in a more direct method showing also spatial inhomogeneity. What could be the reason of the measured spatial uniformity in the present work?"

The question raised by the referee touches upon a number of subtle issues, which we address separately below. We also would like to thank the referee for drawing the work of Vancso *et al.* [Sci. Rep. 6, 29726 (2016)] to our attention.

Firstly, our measured LF noise power spectra are spatially uniform in the sense that they are composed of two well-separated Lorentzians irrespective of position (representative spectra shown in Fig. 3b,d). The associated frequencies f_{α} , f_{β} and amplitudes A_{α} , A_{β} exhibit some degree of spread, and clear systematic trends when bias current and temperature are varied (see Figures 3e,f and Supplementary Figures 3 and 4).

The apparent discrepancy between our observations and STM measurements reported in the literature originates primarily from the very different spatial resolutions of the respective techniques: our C-AFM measurements have a probing area defined by the tip radius < 25 nm, whereas STM typically has resolution < 1 nm. We cannot exclude the possibility that our samples exhibit spatial variations in Fermi level similar to those reported by Vancso *et al.* due to variation in interaction strength between MoS₂ and the gold substrate. In that case, due to the large C-AFM probing area, noise spectra acquired in our measurements will represent a spatial average of these different areas, in contrast to the STM measurements of Vancso *et al.* that resulted in atomic resolution images.

Separately, we would expect the properties of our CVD-grown MoS₂ monolayer sample to be significantly closer to those of the exfoliated *synthetic* MoS₂ samples studied by Vancso *et al.*, compared to the exfoliated *natural* MoS₂ studied by McDonnell *et al.* These authors stated as one of

their key results that "variations in the ratio of S/Mo are seen to vary across a single sample by up to 30%, which is direct evidence that the stoichiometry in natural MoS₂ is not uniform." Accordingly, a variety of different defect types exists in natural MoS₂ besides sulfur monovacancies (see extended discussion in added Supplementary Note 11).

We would like to add that in light of work by Vancso *et al.* who visualized the different charge states V_S(0) and V_S(-1) of sulfur monovacancies, we have made the statement regarding the novelty of our experimental observations more precise (i.e., switching of charge states; see page 11).

"Most significantly, our results represent the first experimental observation of charge state switching in sulfur monovacancies in MoS₂, and our work adds significant detail to the understanding of the dynamic behaviour of these defects."

Comment #3: "Related to the Fermi level position; in Fig 2b the electron occupancy is confusing. The figure shows the neutral V_S(0) situation, while the Fermi level is higher than the "e" defect state. It is worth noting that in the case of the charged state V_S(-1) the defect levels are also modified (see ref 13)."

We thank the referee for the helpful comment. We agree that it is preferable to show the charged V_S(-1) defect configuration which is the ground state at the Fermi level realized in our experiment; we have amended Fig. 2b accordingly.

Comment #4: "Usually the C-AFM measurements are sensitive to the AFM tip. How many AFM tip (and sample) was used during the measurements?"

The reviewer is correct that the state of tip can affect the C-AFM results. For this reason, we took precautions to minimize the effect of AFM tip by including 100 MOhm series resistor, thus restricting the current to 100 nA to prevent tip degradation. We have now added the information that for the acquisition of all data presented in this work, we used a single AFM cantilever (Supplementary Note 3).

"This work is based on a total of 285 spectra, acquired at different positions on the same MoS₂ sample, using a single AFM cantilever. Spectra acquired on other MoS₂ samples (not shown in this work) were qualitatively identical."

Reviewer #2

General comments: "The major claims of the paper are the influence of the sulfur vacancies (which seem to be dominant in MoS₂ but without excluding other possibilities) on the charge mobility reduction in MoS₂ and overall, the stability of the first charged state (one trapped electron), and its role in the n-doping of MoS₂. This latter constitutes a real novelty in the field and is probably of high interest."

"The experiments and their corresponding interpretations presented here are rather convincing, in particular with respect to the noise measurements."

We thank the reviewer for this encouraging assessment.

Comment #1: "However, despite a brief explanation in the supplementary material, I think that an STM image of the defect considered here, i.e. the sulfur vacancy, compared to previous experimental determination and/or theoretical modelling would be to my opinion more convincing for the reader."

We interpret the reviewer's question such that he/she would like us to provide evidence that our measurement is indeed probing sulfur vacancies, as opposed to other defect structures. Since the spatial resolution of our C-AFM measurements is determined by the radius ~ 25 nm of the metallized AFM tip, we are not able to resolve the defect structure with our experimental setup.

However, a separate good argument for the expectation that noise spectra are dominated entirely by sulfur vacancies is that the latter are several orders of magnitude more common than other defect types. To support this argument more fully, we have - in response to the referee's question - compiled a comprehensive survey of the literature on the theoretically calculated formation energies and the experimentally observed areal densities of different defect types in MoS₂ (see Supplementary Note 11).

Additionally, previous work from our laboratory (Jeong et al., 2016) has reported STM measurements on MoS₂ samples grown using a closely similar protocol and the same equipment; that work observed a sulfur monovacancy defect density of $(4 \pm 1) * 10^{11}/\text{cm}^2$.

Comment #2: "Indeed, if the case of oxygen can be a priori discarded, many other possibilities exist in terms of defects in a MoS₂ monolayer. For example, Mo substitution of Sulfur can occur, which is more complicated to detect and would create very reactive dangling bonds able to trap electrons. This latter aspect among other defects should also be considered for the global explanation of n-doping in MoS₂ which I think the authors have reduced too quickly to the single case of sulfur vacancy. Therefore, the discussion should be opened by the authors for more general possibilities beyond the present specific case of sulfur vacancy."

We thank the referee for raising the issue of discussing the influence of other possible defects more broadly.

Regarding the specific case of oxygen adatom and substitutional defects in MoS₂, theoretical work has shown that while both of these defects can indeed form (albeit formation is subject to kinetic barriers), neither of these defects leads to the creation of defect states within the energy gap of MoS₂. (See discussion in expanded Supplementary Note 11.)

Regarding other, electrically active types of defects in MoS₂, we have extended Supplementary Note 11 with a detailed survey of the literature on the various point-defect types possible in MoS₂, listing their formation energies, presence of mid-gap states, and experimentally determined areal densities. The literature data available for CVD-MoS₂ supports the notion that sulfur vacancies are indeed the defect type that dominates sample properties.

The case is different for exfoliated *natural* MoS₂, where strong inter- and intra-sample stoichiometry variability has been reported, linked to the presence of defect types other than sulfur vacancies (see Supplementary Note 11). Accordingly, regarding the universally observed n-type behavior, we have revised the manuscript to limit our extrapolation to the case of CVD-MoS₂ where such complications are absent, whereas we are more tentative concerning exfoliated natural MoS₂ (see page 10, and Supplementary Note 14).

“Separately, the presence of sulfur vacancies has been linked to the experimental observations that contact resistances of MoS₂ devices are high due to Fermi level pinning^{1,6-8,22}, and that thin MoS₂ samples almost universally exhibit n-type carrier transport. This attribution to sulfur vacancies should be regarded as tentative in the case of exfoliated natural MoS₂ samples: natural MoS₂ can exhibit variable stoichiometry even within samples, including regions of sulfur excess^{6,28}, as well as a variety of impurity atoms that could lead to doping²⁸ (see Supplementary Notes 11 and 14 for an extended discussion). For CVD-grown MoS₂ monolayers, however, these complications are absent, and the realization of the V_S(-1) state in a significant fraction of the abundant V_S defects would account for all of the anomalies observed in MoS₂ FET devices.”

Reviewer #3

General comments: "The noise experiment is thoroughly conducted and the manuscript is well written."

We thank the reviewer for this positive view.

Comment #1: "Although the work is interesting, it is rather incomplete. Somewhat global conclusions are derived from a minimal amount of experimental results. Whereas all the drawn conclusions seem to be relatively plausible, the present work relies on a single experiment (Fig 3) with a very few parameters to change (i.e. 2 bias currents and a narrow temperature range)."

We would respectfully disagree with this assessment. We performed the experiment presented in this work at 6 separate temperatures, covering the range 150 - 300 K, and varying the bias current over 3 orders in magnitude, between 0.1 - 100 nA (see Supplementary Figures 3 and 4 for a summary). In all, we feel that our observations and conclusions rest on a solid foundation.

We agree with the referee that it is helpful to include an explicit statement on the volume of our data. The total number of spectra acquired in this work is 285, and we have added this information to Supplementary Note 3:

"This work is based on a total of 285 spectra, acquired at different positions on the same MoS₂ sample, using a single AFM cantilever. Spectra acquired on other MoS₂ samples (not shown in this work) were qualitatively identical."

Comment #2: "Moreover, the manuscript does not present a clear direct experiment, unambiguously linking the measured noise spectrum with the proposed electrical state of the vacancies. Thus, the statement 'noise measurements as a direct probe of the defect dynamics of the sulfur monovacancy' remains unproven and far-fetched."

We are unsure how to interpret the reviewer's comment. If he/she proposes that noise spectroscopy is not a direct measurement of the sample's dynamics, we would disagree with that statement. Noise spectra reflect the presence of switching processes that occur between discrete charge states, associated with trapping/detrapping of charge carriers - in other words, noise spectra represent a direct measure of dynamic processes taking place in the sample.

On the other hand, in case that the reviewer contests only the narrower notion that our measured noise spectra reflect specifically the switching behaviour of sulfur monovacancies: we have extended Supplementary Note 11 by a detailed discussion of the defects in MoS₂ that dominate LF noise; in the case of CVD-MoS₂, sulfur vacancies are the predominant type.

Comment #3: "It would be interesting to investigate a different defective state of the sample, for example after inducing additional defects, e.g. by ion irradiation."

We agree with the referee that investigating the dynamics of other types of defects occurring in MoS₂ will be fascinating – we are planning to conduct such work, but it is beyond the scope of the present work. Samples subjected to ion irradiation, as suggested by the referee, are particularly challenging to investigate as many different types of defects can be generated due to the high ion energies involved. Accordingly, such work will require using an STM (instead of C-AFM), because the structure and areal density of defects is unknown after such treatment, unlike the case of pristine MoS₂ dominated by sulfur monovacancies.

Comment #4: "Fig 1S (b, c) is largely inconclusive. It is impossible even to derive the flake thickness out of it. Part (c) doesn't bring any new information."

We thank the referee for pointing out the shortcomings of Supplementary Figure 1; we have modified the figure, adding a trace extracted from the AFM topography image.

Concerning the relevance of AFM phase lag images, the phase lag between the AFM ac-mode excitation and the cantilever response is a measure for the energy dissipated by the cantilever, i.e., friction between the silicon tip and the sample surface. Accordingly, a phase lag image provides information about chemical homogeneity of the scanned surface, which is not apparent from a topography image alone. In our case, taken together, the topography and phase lag images confirm that the MoS₂ surface is homogeneous and free from polymer residues. We have amended Supplementary Note 2 to include this explanation:

"In ac-mode AFM, the phase lag between the sinusoidal excitation and the cantilever response provides a measure for the energy dissipated by the cantilever, i.e., the friction between the silicon tip and the sample surface³. Accordingly, a phase lag image provides information about the chemical homogeneity of the scanned surface. Taken together, our topography and phase lag images confirm that the MoS₂ surface is homogeneous and free from polymer residues. The measured height difference between MoS₂ and the gold substrate is ~ 1.2 nm (Supplementary Fig. 1e), greater than the known interlayer spacing $c = 0.61$ nm of bulk MoS₂, i.e., we observe an apparent base height offset ~ 0.6 nm. Offsets of this magnitude are commonly seen in AFM measurements on surfaces consisting of dissimilar materials, such as MoS₂ deposited on SiO₂ (refs. 4–6)."

Comment #5: "It would be essential to perform conductive AFM mapping on both large and small scales and at least try to derive different electrical states of defects from there. Additionally, it should be compared with the surface potential distribution over the same area, ideally using a calibrated setup, which is relatively easy for the gold substrate."

We agree with the referee that it would be desirable to perform conductive AFM noise spectroscopic mapping. However, due to a variety of technical challenges, this spectroscopic mapping facility is not currently available.

Separately, due to the limited spatial resolution of the AFM tip (radius < 25 nm), the collected current noise signal typically averages over multiple vacancy defects simultaneously, and cannot be assigned to a single defect. More importantly, our work shows clearly that switching between different vacancy charge states (charges 0, -1, -2) occurs at rates of ~ 4 Hz and ~ 1 kHz - given the

much longer acquisition times (50 sec/spectrum), vacancy defects cannot be assigned a static charge state with the available temporal resolution.

Comment #6: "Fig 2: How original is the Figure? It seems that many parts are borrowed from Refs. 9, 12, 13."

Figure 2a,b,c serves to acquaint the reader with background information that is indispensable for an understanding of our experimental findings, without consulting other literatures. We note that we do cite that literature where appropriate. Regarding the role of the figure:

- Fig. 2a introduces the structure of MoS₂ (which we do not attribute to any reference because we consider it to be in the public domain).

- Fig. 2b shows the energy levels of the sulfur vacancy V_S (calculated in references cited), relative to the Fermi energy as is typically realized in our experiment.

- Fig. 2c (showing the stability of V_S charge states as functions of Fermi energy) is the only panel that is adapted from (and explicitly attributed to) earlier work [Noh *et al.* (2014), Komsa *et al.* (2015)] without modification. We have little doubt that this information is highly relevant to the discussion of our experimental results.

Figure 2d, which summarizes our key findings, originates exclusively from our work. (Please also see also comment #7 below.)

We have now clarified the language relevant passages (caption of Fig. 2; page 5, paragraph 1) to be unambiguous in every regard.

“Theoretical studies^{10,11,22} have predicted that depending on the position of the Fermi energy E_F within the energy bandgap of MoS₂, the neutral state V_S(0), the charged state V_S(-1), and possibly V_S(-2) (ref. 21,22) are most stable, while other charge states are unstable due to strong electrostatic repulsion. The relative stability of V_S charge states is shown in the energy diagram in Fig. 2c (adapted from refs. 10,11).”

Comment #7: "Fig. 2d: How were the energy profiles assigned to certain defect states? Is it purely schematics? Additionally, it says: ‘The energy diagram shown in Fig. 2d is consistent with previous theoretical work.’ My understanding that it is built up based on previous works and fits well to the conclusions of the present paper. The authors should be more specific on how Fig 2d was derived."

This comment appears to touch upon a variety of related issues. We attempt to address these separately below.

(a) The energy levels and height of energy barriers shown in Fig. 2d are qualitative, and the text explicitly identifies the diagram as schematic (see beginning of "Discussion" section). We have now added the word "schematic" to the caption of Fig. 2d in order to be unambiguous.

(For completeness, we note that we determine the energy barriers quantitatively rather than only qualitatively - see bounds quoted on page 8, paragraph 2.)

(b) Concerning the question how the diagram was derived - this, too, was explicitly stated in our manuscript ("Discussion" section, paragraph 1). Please see point (c) below for an extended description.

(c) Regarding the question of the originality of the energy diagram in Fig. 2d: it is entirely original, in the sense that it is based completely on our experimental observations. In response to the referee, we list the relevant observations in the following (also added as new Supplementary Note 12):

- In noise spectra acquired anywhere on our MoS₂ monolayer samples, we observe two switching processes (at characteristic frequencies f_α, f_β). From this observation, we infer the presence of 3 different charge configurations.

- From the temperature dependence of f_α, f_β , we calculate the associated energy barriers E_α, E_β (and capture cross sections $\sigma_\alpha, \sigma_\beta$) via Eqn. (2). These energy barriers are schematically indicated in Fig. 2d.

- The inferred capture cross sections differ by about 3 orders of magnitude, $\sigma_\alpha \ll \sigma_\beta$. Since noise spectra are acquired using an electron current, a neutral defect will have a much larger cross section than a negatively charged defect. Moreover, σ_α and σ_β have orders of magnitude typical for negatively charged and neutral defects in SiO₂, respectively. Furthermore, we can exclude the presence of positive charge states, since these are known to have capture cross sections far larger than those observed in our measurements.

Up to this point, we have inferred that fast switching ($f_\beta \sim 1000$ Hz) occurs between a neutral state and a negatively charged state, whereas slow switching ($f_\alpha \sim 4$ Hz) occurs between two negatively charged states. We identify these states with $V_S(0)$, $V_S(-1)$, and $V_S(-2)$.

- We inspect raw noise current traces to infer the duty cycles associated with different charge states, i.e., what fraction of time is spent in the charge state. This information provides a measure of the relative energy levels. For switching on short timescales [~ 1 msec, associated with f_β , see Fig. 3a,b], we do not observe a preferred level, from which we conclude that $V_S(0)$ and $V_S(-1)$ have approximately equal energy levels. In contrast, switching on long time scales [~ 100 msec, associated with f_α , see Supplementary Fig. 2], we observe that current values dwell in the higher current state [i.e., $V_S(-1)$] for a majority of the time; from this we conclude that $V_S(-2)$ is a higher energy state than $V_S(-1)$.

This concludes the observations upon which we base the schematic energy diagram shown in Fig. 2d.

We then proceed to point out that the energy level diagram constructed by us in this way is indeed consistent with the predictions made by previous theoretical work [Noh *et al.* (2014), Komsa *et al.* (2015)] about the relative formation energies of $V_S(0)$, $V_S(-1)$ and $V_S(-2)$.

As noted, we have now added a new SI section (Supplementary Note 12) that describes our reasoning in detail.

Comment #8: "How does the presence of the gold substrate affect the dynamics of the defects (i.e. by modifying E_f in respect to the defect level)?"

This is correct; the gold substrate will stabilize the Fermi level in the band gap close to the conduction band minimum, thereby increasing the stability of $V_S(-1)$ (see "Discussion" section, paragraph 3).

Comment #9: "It would be useful if the authors could elaborate more on relative independence of both frequencies (f_a and f_b) on current."

The relative independence of trapping/de-trapping frequencies implies that the Fermi level of MoS_2 remains largely unmodified as the bias current is increased. This is, to a degree, expected for a tunneling junction under a small voltage bias. In our experimental setup, this small voltage bias condition seems to hold, since the overall resistance is dominated by the 100 M Ω series resistor. For the range of bias currents employed in our work, the maximum voltage bias across the tunneling junction is below 0.3 V. It is possible that the characteristic frequencies will exhibit current dependence under larger voltage bias; however, we restricted the bias current below 100 nA to prevent tip degradation.

Comment #10: "It should be clarified how assignment of certain frequencies/capture cross-sections to specific charge defects was performed."

Please see our response to comment #7 above.

Comment #11: "'two well-separated Lorentzian peaks in measured power spectra': I suppose the authors mean Fig 3 b,d. It is not obvious that the peaks can be accurately fitted by Lorentzians."

The functional form of Eqn. (1) in our manuscript [$1/(1 + (f/f_i)^2)$] represents a Lorentzian function (= Cauchy distribution) centered at $f = 0$, and of width f_i . This functional form is expected for noise spectra in a situation where a single switching process is present. [In situations where (unlike in MoS_2) defects are not well-defined, noise spectra represent a summation over many defect types, typically resulting in a $1/f$ shape (for example, see Sangwan *et al.* Nano Lett. 13, 4351 (2013); ref. 18 in our manuscript.]

In Fig. 3b,d, we fit the functional form of Eqn. (1), comprising two such Lorentzian spectral shapes (and a constant offset), to our experimentally obtained noise spectra. In our own view, the match between spectra and fits is quite remarkable.

We have improved the wording of relevant passages (page 5, paragraph 3; page 7, paragraph 1; caption of Fig. 3) to clarify these points:

"The corresponding noise power spectra $S_I(f)$ [Fig. 3b,d] are described well as sums of two Lorentzian noise functions [Eqn. (1)] with characteristic frequencies f_α and f_β (green and cyan lines, respectively),...."

"We obtain the numerical values of the relevant physical parameters via least-squares fits to the sum of two

Lorentzian functions”

“Measured spectra are well described by Eqn. (1), comprising two Lorentzian functions of widths $f_\alpha = 4$ Hz (green line) and $f_\beta = 1$ kHz (cyan line).”

Comment #12: “Specifically, the high value of cross section $\sigma(0)$ indicates the involvement of a neutral electron trap, VS(0); in contrast, the low value of $\sigma(0)$ is consistent with a transition between two negatively charged states, VS(-1) and VS(-2).”: this statement is rather hand waving. It is surely a possible explanation, but not confirmed by a direct experiment.”

Regarding the assignment of capture cross sections to charge states, please see our response to comment #7 above.

Comment #13: “Also, considerations on work function are again based on possible scenarios and data taken from literature rather than own measurements. There is no direct prove.”

While in our view, making use of data available in the literature is legitimate and time-honoured scientific practice, we do agree with the referee that given the wide range of work functions reported for gold surfaces under a variety of conditions, it is desirable that we measure the work function of our specific gold substrate. We have done so by Kelvin probe, and the measured work function of our gold substrate is ~ 4.8 eV - see Supplementary Note 13 for details.

Additionally, we have also added a more extensive discussion of available literature on the work function of gold surfaces (in vacuum and in ambient) to Supplementary Note 13.

Reviewers' comments:

Reviewer #1 (Remarks to the Author):

I think most of the points raised by the reviewers have been addressed satisfactorily. I would like to note that the Fermi level position and its effect on the charge configurations (0,-1,-2) could have been investigated in more detail by using a gate electrode in the experiments. In this case the authors could avoid the "speculative extrapolation" part in the discussion. Nevertheless, the present form of the manuscript is suitable for publication in Nature Communications.

Reviewer #2 (Remarks to the Author):

The authors have satisfactorily answered to my comments, providing a strong argumentation to support their hypothesis on the S-vacancy. They have also discussed the possibilities of other defects, which makes the article more convincing. Consequently, I recommend this work for publication in Nature Communications.

Reviewer #3 (Remarks to the Author):

Although the technical quality of the main manuscript and, in particular, the Supplementary Information, was significantly improved after revision, I am still not convinced that the manuscript is merited to be published in the Nature Comms. The main issues are on a fundamental level and reflect the limited methodological approach of the authors. Unfortunately, the manuscript still remains based on a single experimental technique - although thoroughly conducted, it does not have capability to present unambiguous direct information about the phenomena in question. The main conclusions are still largely speculative, no direct proves are presented.

Comment 1: We would respectfully disagree with this assessment. We performed the experiment presented in this work at 6 separate temperatures, covering the range 150 - 300 K, and varying the bias current over 3 orders in magnitude, between 0.1 - 100 nA (see Supplementary Figures 3 and 4 for a summary). In all, we feel that our observations and conclusions rest on a solid foundation.

We agree with the referee that it is helpful to include an explicit statement on the volume of our data. The total number of spectra acquired in this work is 285, and we have added this information to Supplementary Note 3

I still insist that the temperature range is narrow, but even so, the authors do not use it meaningfully, for example by extracting the relevant change in the activation energies, when they discuss the tunnelling barriers.

By no means have I suggested to increase the volume of noise data presented. I am pretty sure that this data is alright. My statement is related to necessity of data obtained by COMPLIMENTARY methods, which would allow to move from a speculative approach to confirmed, solidly proven conclusions.

Comment 2 (see also Review 2/comment 2): still remains vague. All other possible defects are ruled out on the base of theoretical works and literature reviews. Nothing is wrong in using both of them, of course. However, it seems that the authors pick up 2 possible defects, which fit nicely to their model of universe, the rest is simply discharged.

This comment is also linked to the question on a strong necessity of defect mapping either by STM (Review 2/comment 1) or alternatively TEM and/or small scale c-AFM (my comment 5). Answers to both questions are unsatisfactory: 1) I do not see any previous STM studies in the work performed by Jeong (2016) [Ref 13 I believe?]; 2) I read the answer that scanning in the c-AFM mode is not possible, which is rather strange and further limits the capability of the method; 3) an ideal prove would be a consequent imaging of a same small area or part of it by both techniques and

performing the noise spectroscopy on the known area.

Comment 3: '...because the structure and areal density of defects is unknown after such treatment, unlike the case of pristine MoS₂ dominated by sulfur monovacancies.'

Domination of monovacancies in CVD MoS₂ is still not confirmed.

Comment 4: '...the phase lag between the AFM ac-mode excitation and the cantilever response is a measure for the energy dissipated by the cantilever, i.e., friction between the silicon tip and the sample surface. Accordingly, a phase lag image provides information about chemical homogeneity of the scanned surface, which is not apparent from a topography image alone.

The AFM phase image is an INDIRECT measure of energy dissipation. It is NOT a measure of mechanical friction and it does NOT in general provide information about chemical inhomogeneity of the sample (however, in some cases certain information can be indirectly (!) extracted, but only in the case of very carefully conducted experiments, which I do not see here).

We thank the referees for their time and effort in reviewing our manuscript, and for helpful criticism that has prompted us to improve our manuscript in numerous ways for the revised version that we resubmitted for review recently.

To our great regret, even though reviewers #1 and #2 were supportive of publication, Nature Communications decided to reject our revised manuscript based on the negative opinion of reviewer #3.

We welcome informed and constructive criticism of our work. What we do find highly problematic, however, is that the criticisms voiced by reviewer #3 regarding our revised manuscript are in their majority factually wrong, besides being unconstructive and unhelpful. It is clear from his/her comments that reviewer #3 has no significant expertise in the fields most relevant to our work, namely in the areas of low-frequency noise studies and defects in MoS₂, and thus it is difficult to see how he/she can assess our work in a meaningful way. Accordingly, we decided to appeal the decision of Nature Communications, and we are resubmitting our manuscript without further changes.

Below, we provide a point-by-point discussion of all reviewers' comments.

Reviewer #1

Reviewer's comment: "I think most of the points raised by the reviewers have been addressed satisfactorily. I would like to note that the Fermi level position and its effect on the charge configurations (0,-1,-2) could have been investigated in more detail by using a gate electrode in the experiments. In this case the authors could avoid the "speculative extrapolation" part in the discussion. Nevertheless, the present form of the manuscript is suitable for publication in Nature Communications."

We thank the reviewer for his/her support of our work, and for previous insightful comments. We agree with the reviewer that integration of a gate electrode in our experimental apparatus would greatly expand the capability of the technique, and in fact such an extension is planned for our future work.

Reviewer #2

Reviewer's comment: "The authors have satisfactorily answered to my comments, providing a strong argumentation to support their hypothesis on the S-vacancy. They have also discussed the possibilities of other defects, which makes the article more convincing. Consequently, I recommend this work for publication in Nature Communications."

We thank reviewer #2 for this positive assessment of our work. We would also like to take the opportunity to credit the reviewer for comments in his/her initial referee report that prompted

us to include a much more extensive discussion of defect types present in MoS₂ in our revised manuscript, to more fully support the notion that sulfur monovacancies are the dominant defect type in MoS₂.

Reviewer #3

Comment #A: "Although the technical quality of the main manuscript and, in particular, the Supplementary Information, was significantly improved after revision, I am still not convinced that the manuscript is merited to be published in the Nature Comms. The main issues are on a fundamental level and reflect the limited methodological approach of the authors. Unfortunately, the manuscript still remains based on a single experimental technique - although thoroughly conducted, it does not have capability to present unambiguous direct information about the phenomena in question. The main conclusions are still largely speculative, no direct proves are presented."

Here the reviewer asserts that noise spectra do not represent a *direct* measure of defects' charge state dynamics.

We already responded to the same criticism in our earlier response letter. In short, the reviewer's statement is simply wrong: current switching seen in noise current spectra is a *direct* reflection of the switching processes that occur between discrete charge states, associated with trapping/detrapping of charge carriers - i.e., noise spectra represent a *direct* measure of dynamic processes taking place in the sample.

To quote a key review paper in the field of low-frequency noise studies, we add a passage from Kirton and Uren [*Adv. Phys.* 38, 367 (1989)]:

"[in a graph of] the random telegraph signal (RTS) measured in the drain current of a MOSFET as a function of time, the times in the high- and low-current states correspond to carrier capture and emission respectively."

Comment #B: "I still insist that the temperature range is narrow, but even so, the authors do not use it meaningfully, for example by extracting the relevant change in the activation energies, when they discuss the tunnelling barriers."

In this comment, reviewer #3 suggests that we should have evaluated the energy barrier height between the 0, -1, -2 charge states of V_S *as a function of temperature*.

It appears that the reviewer misunderstood how the energy barriers are derived from our experimental data. We perform this evaluation using Eq. (2) [page 7 of our manuscript]. For the purpose of this argument, we rewrite Eq. (2) by combining all prefactors into a constant C , such that the equation reads $f(T) = C \exp(-E/k_B T)$, where E is the energy barrier height, and C contains the scattering cross section. Importantly, *both* C and E are unknown and have to be evaluated from the *same* data set; to facilitate this evaluation, the activation energy E is

assumed to be a *constant*, independent of temperature.

We evaluate C and E by plotting $\log(f)$ vs $1/T$ and performing a linear fit: we obtain the value of the energy barrier E from the fit slope, and C from the extrapolated f -axis intercept [see description in our manuscript (page 8) and Fig. 3e,f]. We note that the observed linear relationship between $\log(f)$ and $1/T$ confirms that E is indeed temperature-independent within the accuracy of our data.

While it is conceivable that the activation barrier E might vary as a function of temperature, such a variation cannot be explored with the currently available data set and the referee's suggestion is simply not feasible.

Comment #C: "By no means have I suggested to increase the volume of noise data presented. I am pretty sure that this data is alright. My statement is related to necessity of data obtained by COMPLIMENTARY methods, which would allow to move from a speculative approach to confirmed, solidly proven conclusions."

In his/her previous referee report, reviewer #3 asserted that we has presented "a minimal amount of experimental results" and "only 2 bias currents". We are glad to discover that the reviewer has changed his/her mind in that regard and now agrees that we have provided an adequate amount of noise spectroscopy data in our manuscript.

Regarding complimentary methods to verify the dominance of sulfur vacancy defects in MoS_2 , our revised manuscript includes a comprehensive review of the literature that fully establishes this point - please see comment #D below.

Comment #D: "Comment 2 (see also Review 2/comment 2): still remains vague. All other possible defects are ruled out on the base of theoretical works and literature reviews. Nothing is wrong in using both of them, of course. However, it seems that the authors pick up 2 possible defects, which fit nicely to their model of universe, the rest is simply discharged."

In this comment, the reviewer appears to suggest that among a large number of equally possible defects of MoS_2 monolayers, we arbitrarily select sulfur monovacancies (V_S) and divacancies (V_{S_2}) to assign our experimental observations to, and equally arbitrarily exclude all other defects from consideration.

This accusation has absolutely no basis. In Supplementary Note 11, we discuss the prevalence and electrical activity of point-like defects in MoS_2 at great length. We gather that this 7-page-long Supplementary Note has escaped the notice of reviewer #3; otherwise we cannot explain this ignorant and spiteful comment.

In contrast to the reviewer's claim, there is nothing vague or arbitrary about our discussion of the MoS_2 defect types. In our revised manuscript, we have greatly extended Supplementary Note 11 (see pages 12-18) to present a comprehensive overview of existing literature on point-like defects in MoS_2 , covering both theoretical calculations and experimental observations by STM and TEM. The consensus of all that work is that sulfur monovacancies (V_S) are by far the most common point-like defects in exfoliated and CVD- MoS_2 , followed

by divacancies (V_{S_2}) [see in particular Supplementary Tables 1 to 3, listing 13 literature references the content of which is discussed in detail].

In sum, we have assigned our observations to the charge state dynamics of sulfur monovacancies (V_S) because previous theoretical and experimental work has presented overwhelming evidence that amongst the electrically active defect types present in MoS_2 , V_S is the most prevalent defect type by far.

Comment #E: "This comment is also linked to the question on a strong necessity of defect mapping either by STM (Review 2/comment 1) or alternatively TEM and/or small scale c-AFM (my comment 5). Answers to both questions are unsatisfactory: 1) I do not see any previous STM studies in the work performed by Jeong (2016) [Ref 13 I believe?]"

Here, reviewer #3 correctly highlights the importance of characterizing point-like defects in MoS_2 by high-resolution probes, such as STM and TEM. Indeed, our institute has performed and published STM measurements on our MoS_2 material.

We pointed out that fact in our previous response letter to the reviewers: "Additionally, previous work from our laboratory (Jeong et al., 2016) has reported STM measurements on MoS_2 samples grown using a closely similar protocol and the same equipment; that work observed a sulfur monovacancy defect density of $(4 \pm 1) \times 10^{11}/cm^2$."

The referenced work from our laboratory [Ref. 13] is Jeong *et al.* *ACS Nano* 10, 770 (2016). Its relevancy for our present manuscript is that it established the areal density of sulfur vacancy defects for our MoS_2 material [$(4 \pm 1) \times 10^{11} cm^{-2}$].

The STM data in Ref. 13 that reviewer #3 was unable to locate are presented in Fig. S8 (in the Supplementary Information). Below, we reproduce the scanning tunneling spectra shown in Fig. S8, which support the notion that the defects that were studied in Ref. 13 are sulfur vacancies:

The inset shows STM topography data, highlighting a sulfur vacancy (blue circle). In STS data acquired at this position, additional states introduced by the sulfur vacancy are apparent, close to the conduction band minimum, consistent with theoretical work. The density of vacancy defects that was determined in that work by STM [$(4 \pm 1) \times 10^{11} cm^{-2}$] is stated in the caption of Fig. S8, along with relevant technical information.

Comment #F: "2) I read the answer that scanning in the c-AFM mode is not possible, which is rather strange and further limits the capability of the method"

What is even stranger is how reviewer #3 twists our words. In our response letter to the reviewers, we wrote: "We agree with the referee that it would be desirable to perform conductive AFM noise spectroscopic mapping. However, due to a variety of technical challenges, this spectroscopic mapping facility is not currently available."

The technical challenges to noise spectroscopic mapping mentioned in our statement are limitations of the hardware and software of the C-AFM instrument available to us at present [Hitachi E-Sweep AFM]. *In no way* did we imply that there are fundamental impediments that make noise spectroscopic mapping impossible, and to misrepresent our statement to this effect suggests substantial ill will on the part of reviewer #3.

Comment #G: "3) an ideal prove would be a consequent imaging of a same small area or part of it by both techniques and performing the noise spectroscopy on the known area."

While the reviewer's suggestion may or may not constitute an ideal proof, we have instead chosen the approach of compiling a review of the large body of literature that has been published on the topic of point-like defects in MoS₂ (including work from our own institute on MoS₂ samples closely similar to those used in our present manuscript), to establish that the sulfur monovacancy is the dominant defect type (see comment #D above).

Comment #H: "Domination of monovacancies in CVD MoS₂ is still not confirmed."

While reviewer #3 adopts an authoritative tone here, his/her statement is nonetheless factually wrong, regardless of how often he/she repeats it - it is well-known in the research field that sulfur monovacancies (V_S) are the dominant defect type in CVD-MoS₂. (Please see comment #D above.)

Comment #I: "Comment 4: '...the phase lag between the AFM ac-mode excitation and the cantilever response is a measure for the energy dissipated by the cantilever, i.e., friction between the silicon tip and the sample surface. Accordingly, a phase lag image provides information about chemical homogeneity of the scanned surface, which is not apparent from a topography image alone."

The AFM phase image is an INDIRECT measure of energy dissipation. It is NOT a measure of mechanical friction and it does NOT in general provide information about chemical inhomogeneity of the sample (however, in some cases certain information can be indirectly (!) extracted, but only in the case of very carefully conducted experiments, which I do not see here)."

Here, the reviewer #3 refers to our Supplementary Note 2 (Sample characterization).

We are a bit puzzled why the reviewer elected to attack this particular passage of our

manuscript with such fervour: in the offending paragraph quoted by the reviewer, we presented general sample characterizations (Raman spectra, AFM topography and phase lag images) in order to establish that our sample is monolayer MoS₂ and free from polymer contamination. In contrast, the passage does not contain any claims concerning the key topic of our manuscript, i.e., sulfur vacancies and their properties - in his/her eagerness to attack our work, reviewer #3 seems to have missed that point.

As for the content of his/her criticism, the reviewer writes in a rather categorical manner, but most statements are simply wrong: AFM phase lag images *do* provide information on friction between the sample surface and cantilever tip, and in this fashion they *can* provide information on the chemical homogeneity of the sample surface. The relationship between phase lag and surface chemistry is well-established and the utility of phase lag image to probe surface homogeneity (from which we showed that our sample surface is free of contamination) is undisputed.

To counter reviewer #3's uniformed claim, we quote from the pioneering work of Charles Lieber's group [Noy *et al.*, *Langmuir* **14**, 1508 (1998)], which is aptly titled "Chemically-Sensitive Imaging in Tapping Mode by Chemical Force Microscopy: Relation between Phase Lag and Adhesion":

“These data show that the phase contrast between chemically distinct monolayer regions correlates directly with adhesion forces between the tip and sample in these different regions. In addition, fitting these data to a driven oscillator model shows that differences in phase shift between distinct regions of the patterned SAMs can be quantitatively related to differences in the work of adhesion, W_{st} . Because adhesion forces are readily interpretable on the basis of surface chemical functionality, these studies demonstrate that tapping mode CFM can be used to image samples with chemical sensitivity”

The comprehensive review of Garcia & Perez [*Surf. Sci. Rep.* **47**, 197 (2002)] provides many more examples of using phase lag imaging to study chemically inhomogeneous samples. Regarding the utility of phase lag data, that work states:

“The usefulness of phase imaging to record material properties variations in heterogeneous samples or enhance topographic contrast in samples with large topographic variations is beyond doubt.”

REVIEWERS' COMMENTS:

Reviewer #2 (Remarks to the Author):

I have read throughout the different comments from the Reviewer 3 as well as the corresponding answers from the authors, and I still think that this article deserves to be published in Nature Communications. My main point of disagreement with the authors was related initially to the nature of the defect, and as pointed by Reviewer 3, I asked for a complementary determination like STM image. However, it seems that this feature is not available for now, and in addition, the supplementary informations provided by the authors through an extensive literature analysis give enough information to support the S-vacancy nature of the defect. Hence, if several complementary methods often constitute a full proof of what the authors want to determine, I think that in the present case, the confrontation with literature gives enough information to assert the nature of the defect.

Consequently, I maintain my opinion on this work, which undoubtedly deserves to be published in Nature Communication.

Response to referee comments

Reviewer #2 (Remarks to the Author): I have read throughout the different comments from the Reviewer 3 as well as the corresponding answers from the authors, and I still think that this article deserves to be published in Nature Communications. My main point of disagreement with the authors was related initially to the nature of the defect, and as pointed by Reviewer 3, I asked for a complementary determination like STM image. However, it seems that this feature is not available for now, and in addition, the supplementary informations provided by the authors through an extensive literature analysis give enough information to support the S-vacancy nature of the defect. Hence, if several complementary methods often constitute a full proof of what the authors want to determine, I think that in the present case, the confrontation with literature gives enough information to assert the nature of the defect. Consequently, I maintain my opinion on this work, which undoubtedly deserves to be published in Nature Communication.

We would like thank the referee for the time and effort spent in reviewing our manuscript, and for his/her support of our work. Besides, we also want to thank the referee for previous critical comments that we have found very helpful in improving our manuscript.